# Pro-Degradant Activity of Naturally Occurring Compounds on Polyethylene in Accelerate Weathering Conditions

**DOI:** 10.3390/ma12010195

**Published:** 2019-01-08

**Authors:** Nadka Tzankova Dintcheva, Delia Gennaro, Rosalia Teresi, Marilena Baiamonte

**Affiliations:** Dipartimento di Ingegneria Civile, Ambientale, Aerospaziale, dei Materiali, Università di Palermo, Viale delle Scienze, Ed. 6, 90128 Palermo, Italy; deliagen@hotmail.com (D.G.); rosalia.teresi@unipa.it (R.T.); marilena.baiamonte@unipa.it (M.B.)

**Keywords:** polyethylene, naturally occurring compounds, pro-oxidant activity, photo-oxidation

## Abstract

In this work, naturally occurring compounds, such as Vitamin E (VE) and Ferulic Acid (FA), at high concentrations, have been considered as pro-degradant agents for Low Density Polyethylene (PE). However, all obtained results using the naturally occurring molecules as pro-oxidant agents for PE have been compared with the results achieved using a classical pro-oxidant agent, such as calcium stearate (Ca stearate) and with neat PE. The preliminary characterization, through rheological, mechanical and thermal analysis, of the PE-based systems highlights that the used naturally occurring molecules are able to exert a slight plasticizing action on PE and subsequently the PE rigidity and crystallinity slightly decrease, while the ductility increases. To assess the pro-degradant activity of the considered naturally occurring compounds, thin films of neat PE and PE-based systems containing 2 and 3 wt.% Ca stearate, VE and FA have been produced and subjected to accelerated weathering upon UVB light exposure. All obtained results point out that the VE and FA, at these high concentrations, exert a clear pro-oxidant activity in PE and this pro-oxidant activity is very similar to that exerted by Ca stearate. Moreover, the VE and FA at high concentrations can be considered as suitable eco-friendly pro-degradant additives for PE, also in order to control the polyolefin degradation times.

## 1. Introduction

Polyethylene (PE) is the thermoplastic most commonly used for manufacturing of films, sheets, containers, pipes, cable jacketing, etc. for various kinds of food and industrial packaging, the piping industry, and the agricultural field. The PE has found numerous applications because of its low cost and good intrinsic properties and performance, such as exceptional ductility, chemical resistance, humidity barrier properties, etc. Currently, the worldwide production of polyethylene is significantly increasing due to increasing industrial requests, versatility and utility, even if there are significant environmental problems related to its reuse, recycling, disposal and/or energy recovery [1].

Generally, the degradation and biodegradation of the polymers can occur through two different mechanisms, which are hydrolysis and oxidation. As is well known, the polyolefins degrade mainly following the oxidation mechanism and there are the bases of term oxo-biodegradable polyolefins [2]. However, it is generally recognized that the polyolefins, including the polyethylene, are hydrophobic, bioinert and highly resistant to assimilation by microorganisms, such as fungi and bacteria, and their degradation and biodegradation proceed very slowly because of their high molecular weights. Nevertheless, it has been demonstrated that the polyethylene, like all kinds of polyolefins, cannot biodegrade since the molecular weight must be less than 500 g/mol for this to occur [3]. Besides, the polar polyolefin fragments are biodegradable due to their reduced molar mass and incorporated polar oxygen-containing groups, such as acids, alcohols and ketones, which causes microorganism attack acceleration [2,3,4]. Nonetheless, the degradation of the polyolefins is known as oxo-biodegradation, which consists of a two-stage sequential process: first, oxidative degradation, which is abiotic, and second, biodegradation of the oxidation products [5,6,7].

However, there are different approaches to gain and control the environmental biodegradation of polyethylene by the introduction of: (i) transition metal complexes (mainly iron and calcium—organic complexes); (ii) photo-sensible additives and pigments; (iii) starch-based constituents [2,3,4,5,6,7,8,9,10]. It is worth noting that all these additives are able to accelerate the oxidative degradation of the polyethylene, contributing to increase the number of oxygen-containing groups in the first stage of its oxo-biodegradation process. Furthermore, the abiotic pro-oxidation process is the rate determining step for the polyolefin oxo-biodegradation [6,7]. On the other hand, all documented results obtained following all these approaches pointed out a considerable contrasting effect with the stabilizing molecules, usually used to ensure unchanged performance of the polyolefins manufactures during their service life. Also, some of these additives are not eco-friendly.

However, it has been demonstrated by our research group that some natural phenolic compounds are able to exert concentration anti-/pro-oxidant activity in biopolyesters, such as polylactic acid, PLA [11,12]. In particular, if the natural compounds are added at low concentration to the biopolyester, from around 0.1 wt.% to 0.5 wt.%, they are able to protect the PLA against the oxidative degradation, but if they are added in large amounts, from around 2 wt.% to 3 wt.%, the PLA undergoes faster hydrolytic, photo- and thermo-degradation than the neat matrix [13]. The latter issue opens new windows into the use of some natural compounds as pro-oxidant additives and the control of degradation time of the biopolymers. Therefore, the possibility to induce abiotic pro-oxidation of biopolymer and polymers through the introduction of eco-sustainable naturally occurring molecules, replacing the classical synthetic ones, can be considered an advanced matter in this research.

Based on the above-mentioned considerations, in this work, we propose an innovative approach to gain and control the time of the abiotic pro-oxidation process of polyethylene, upon UVB exposure, through the introduction of large amounts (2 wt.% and 3 wt.%) of eco-friendly and non-toxic natural occurring compounds, such as Vitamin E (VE) and Ferulic Acid (FA). In addition, their pro-degradant activity has been compared to that of the calcium stearate (Ca stearate), which are classical pro-oxidant agents for polyolefin.

## 2. Materials and Methods

### 2.1. Materials

The used materials are:-Low Density PolyEthylene, PE, (Riblene^®^ FC30, from Versalis spa, Mantova, Italy), with Mw = 175.000 g mol^−1^, Mw/Mn = 5.76, melt flow index (ASTM D 1238) 0.27 dg/min, density 0.922 g/cm^3^, melting temperature 113 °C;-Octadecanoic acid calcium salt, named calcium stearate (Ca stearate), has been purchased by Sigma-Aldrich Chemie GmbH, Steinheim, Germany, 6.6–7.4% Ca basis.-2,5,7,8-Tetramethyl-2-(4′,8′,12′-trimethyltridecyl)-6-chromanol,5,7,8-Trimethyltocol, named Vitamin E (VE), has been purchased by Sigma-Aldrich. Molecular Weight 430.71.-4-Hydroxy-3-methoxycinnamic acid, named Ferulic Acid (FA), has been purchased by Sigma-Aldrich. Molecular Weight 194.18.

### 2.2. Processing

The preparation of PE-based samples was carried out using a Brabender mixer at T = 180 °C and mixing speed 50 rpm for 5 min. The compounds, such as Ca stearate, VE and FA, have been added at 2 wt.% and 3 wt.%. The neat PE matrix has been subjected to the same processing conditions. Thin films (thickness about 100 μm) of neat PE and all systems containing different compounds have been obtained through compression molding in a Carver Laboratory Press at a pressure P = 1500 psi for 5 min and at T = 180 °C.

### 2.3. Characterizations

Rheological tests were performed using a strain-controlled rheometer (model ARES G2 by TA Instrument, New Castle, DE, USA) in parallel plate geometry (plate diameter 25 mm). The storage (G′) and loss (G′′) moduli were measured performing frequency scans from ω = 10^−1^ to 102 rad/s at T = 180 °C. The strain amplitude was γ = 2%, which preliminary strain sweep experiments proved to be low enough to be in the linear viscoelastic regime.

The calorimetric data were evaluated by differential scanning calorimetry (DSC) using a calorimeter (Perkin-Elmer DSC7, Waltham, MA, USA). All experiment were performed under dry N_2_ on samples of around 10 mg in 40 μL sealed aluminum pans. Four calorimetric (two heating: 30–220 °C and two cooling: 220–30 °C) scans were performed for each sample at scanning heating/cooling rate of 5 °C/min.

Mechanical tests were carried out according to American Standard Test Method - ASTM test method D882 by using an Instron machine mod. 3365. The samples, stored for one week at room temperature and humidity, were tested at 1 mm/min up to a strain of 10%; then, the speed was increased up to 100 mm/min until break. Young’s modulus, tensile strength and elongation at break were recorded, and the data reported represent the average values obtained by analyzing the results of eight tests per sample; the variability of mechanical tests was typically of order of ±5%.

Fourier transform infra-red (FT-IR) spectra were evaluated using the Spectrum One Spectrometer and its Spectrum software (model Spectrum One, Perkin-Elmer, Shelton, CT, USA). The spectra were obtained using 16 scans and a 4 cm^−1^ resolution. The variations of the carbonyl band areas were determined from peak absorption area between 1850–1650 cm^−1^ and reference peak area measured between 2110–1980 cm^−1^). Measurements, carried out in atmosphere oxygen and humidity conditions, were obtained from the average of triplicate samples. 

### 2.4. Accelerated Weathering

The artificial accelerated photo-oxidation tests were performed using a Q-UV chamber mounting eight UV-B lamps. The weathering conditions, in the presence of oxygen, were 8 h of light at T = 55 °C and 4 h of condensation at T = 35 °C. 

## 3. Results

### 3.1. Characterization of PE-Based Systems Containing Naturally Occurring Compounds

To investigate the effect of the presence of large amounts of considered naturally occurring compounds on the performance and properties of PE, accurate preliminary characterization, through rheological, mechanical and thermal analysis, of all PE-based systems has been performed.

In Figure 1, the trends of the torque values recorded during the melt processing on neat PE and PE-based systems containing 2 and 3 wt.% of Ca stearate, VE and FA are reported. It is worth noting that the torque values of neat PE slightly decrease during the two minutes of the processing and remain almost unchanged after that, pointing out that the PE does not undergo any thermo-mechanical degradation during the considered processing times. The adding of 2 wt.% and 3 wt.% of Ca stearate does not influence the trends of the torques, see Figure 1a, signifying no plasticizing effect of the Ca stearate on PE at these concentrations. No plasticizing effect on the PE has been observed also by the presence of 2 wt.% and 3 wt.% of the considered naturally occurring molecules, see Figure 1b,c. As is known, the plasticizing action of low molecular weight molecules, like naturally occurring compounds, can be attributed to their ability to create free volume in the systems by introducing themselves between the polymer chains. The well pronounced plasticizing action of some natural phenolic compounds has been clearly observed and documented for polylactic acid based systems containing 2 wt.% and 3 wt.% [13], while the current obtained results for PE-based systems highlight no plasticizing effect of the considered low molecular weight molecules, probably because of high intrinsic PE viscosity.

In addition, the rheological behavior in oscillatory test of neat PE and PE-based systems containing Ca stearate, VE and FA has been studied and the Figure 2a–c, the values of storage (G′) and loss (G′′) moduli of the investigated systems as a function of the frequency are shown. As expected, neat PE shows high values of both moduli in the whole investigated frequency range. The trends of both G′ and G′′ moduli for all investigated PE-based systems containing Ca stearate and naturally occurring molecules are slightly lower than that observed for the neat PE. Overall, the trends of the storage (G′) and loss (G′′) moduli remains almost unchanged upon the adding of the additives, probably because of high intrinsic PE viscosity and inability of Ca stearate, VE and FA, at these concentrations in order to create additional free volume in the systems.

To estimate the effect of the added compounds on the PE properties in the solid state, mechanical tensile tests have been carried out and the obtained values of the elastic modulus (E), tensile strength (TS) and elongation at break (EB) are reported in Table 1. Generally, it can be noticed that the elastic modulus values slightly decrease and tensile strength values remain almost unchanged, while the elongation at break slightly increases due to the presence of all considered additives. These results can be understood considering the presence of Ca stearate, VE and FA molecules facilitate the slipping the molecules during the tensile test. 

Moreover, the thermal properties of all considered samples has been investigated and the results regarding the fusion and crystallization enthalpies are reported in the last two columns in Table 1. The temperatures of the fusion and cooling peaks, not reported here for brevity, are almost unchanged by the presence of the low molecular weight molecules, while the fusion and crystallization enthalpies slightly decrease and in doing so highlight a slightly pronounced plasticizing effect.

It can be summarized that the considered natural compounds, i.e., VE and FA, exert a slightly pronounced plasticizing effect on PE, very similar to that exerted by the Ca stearate, which is a classical pro-oxidant additive. However, it is very important to highlight that the obtained morphological changes in PE upon the adding of Ca stearate, VE and FA occur to the same extent, pointing out that there is no influence of the morphology changes of the PE-based systems on their photo-oxidation behavior.

### 3.2. Pro-Oxidant Activity of Naturally Occurring Compounds in PE-Bases Systems in Accelerated Weathering Conditions

The ability of VE and FA to exert abiotic pro-oxidant effect in accelerated weathering conditions has been evaluated subjecting PE-based thin films to UVB exposure and the obtained results have been compared to that of neat PE and PE/Ca stearate systems. According to the literature, the photo-oxidation degradation of PE proceeds with: (i) overall accumulation of oxygen-containing groups, such as carboxylic acids, ketones, esters and lactones; (ii) formation of –OH groups, coming from the decomposition of the intermediate alkyl-hydroperoxides; (iii) formation of insaturations, due to disproportion and chain scission [2,7,14]. In this work, the oxidation of neat PE and PE-based systems has been followed monitoring the overall formation of oxygen-containing groups in the carbonyl region, i.e., 1850–1650 cm^−1^, and the ductility loss, e.g., elongation at break trend as a function of the weathering time.

In Figure 3, the FTIR spectra of neat PE and PE-based systems containing Ca stearate, VE and FA at 2 and 3 wt.% at different exposure times are plotted. It can be noted that the UVB exposure of neat PE leads to arising of main complex peak in the region 1850–1650 cm^−1^, due to the presence of >C(=O) groups, and other two smaller peaks in the regions 3600–3300 cm^−1^ and 1250–1170 cm^−1^, due to the presence of –OH groups and ether-type linkages, respectively, see Figure 3a. In the spectra of PE-based systems containing Ca stearate, see Figure 3b,c, the growth of complex carbonyl peak, i.e., in the region 1850 –1650 cm^−1^, can be clearly observed, while the other two peaks are less pronounced in comparison to the neat PE. Furthermore, the presence of classical pro-oxidant agent, such as Ca stearate, leads to a very fast formation of oxygen-containing groups, promoting in this way the abiotic pro-oxidant effect. The same considerations can be made for the pro-oxidant activity of VE and FA in PE, see Figure 3d–g, where the spectra of PE-based systems containing VE and FA are shown. However, it is worth noting that the investigated samples have been subjected to UVB exposure for different time intervals because they become brittle at different weathering times. 

To quantify the accumulation of the oxygen-containing groups in time, the carbonyl index for all investigated samples has been calculated as a ratio between the area of the complex peak in the carbonyl region 1850–1650 cm^−1^, at different exposure times, and the area of the reference peak in the region 2110–1980 cm^−1^, which is referred to the bending vibration of the -CH_3_ [15]. In Figure 4, the carbonyl index for neat PE and all PE-based systems as a function of the weathering time is plotted.

As expected, the carbonyl accumulation for the PE samples containing Ca stearate, at both considered concentrations, show a rapid accumulation of the oxygen-containing groups with respect to the neat PE, see Figure 4a. Interestingly, also the adding of 2 wt.% and 3 wt.% of VE and FA leads to a rapid accumulation of the oxygen-containing groups, as noticeable in Figure 4b,c, and the growth of the carbonyl index for these samples is very similar to those observed for the PE/Ca stearate samples. Besides, the VE molecules are able to efficiently accelerate the oxidation progress, also in the early stage of the UVB exposure, i.e., no induction period for the oxidation process can be observed. The ability of VE molecules to promote the formation of oxygen-containing groups in PE, also in the early stage of the weathering, can be understood considering that the VE molecules form tocopherol radicals upon UVB light, and these radicals processes the peroxidation by themselves, in agreement with the literature [15]. Further, the FA molecules at both considered concentrations are also able to accelerate the formation of oxygen-containing groups in PE matrix in a similar way to the Ca stearate.

Furthermore, to evaluate the ductility loss of the PE-based systems containing pro-oxidant agents, the elongation at break, EB, as a function of the weathering time has been monitored and in Figure 5, the trends of the dimensionless EB are plotted. The dimensionless EB values have been calculated as ratio between the EB values at given exposure time, EB(t), and the EB values before the photo-oxidation, EB(t_0_). As is known, the ductility loss of the polyolefins during the degradation process can be attributed to two phenomena related among them, specifically, the formation of new chemical groups, due to the accumulation of oxygen-containing groups, and the reduction of polymer molecular weight, because of the fragmentation of the polymer chains [2,7,14]. 

Therefore, the presence of Ca stearate at both considered concentrations leads to a more rapid reduction of the dimensionless EB, which means a more rapid ductility loss in comparison to the neat PE, see Figure 5a. It is interesting to highlight that the presence of VE and FA also causes more rapid decreases of the dimensionless EB, i.e., ductility loss, with respect to the neat PE. 

Hence, it is important to highlight that the ductility loss of all investigated samples upon UVB exposure occurs more rapidly than the accumulation of the oxygen-containing groups, and for this reason, in Figure 4 and Figure 5, the weathering time intervals are different. Besides, the Ca stearate promotes the abiotic oxidation of PE upon UVB light, but it is worth noting that the samples become brittle, even if the accumulation of carbonyl species has no high values; this effect is more pronounced at high concentration. Exactly the same considerations can be made for the activity exacerbated by the FA molecules in PE. The samples containing VE molecules become brittle at longer exposure time in comparison to that containing FA molecules and they clearly promote the accumulation of oxygen-containing groups.

Finally, all obtained results point out the ability of VE and FA molecules at high weight concentrations to promote the abiotic pro-oxidation in a similar way to a classical pro-oxidant agent, such as Ca stearate. This result could be explained considering that in a very short exposure time, the FA and VE molecules are able donate hydrogen atoms, but at a prolonged exposure time, presumably due to the presence of very large number of ferulic and tocopherol radicals, the hydrogen tearing from the macromolecules can be considered to be a predominant effect, leading to the formation of large number of intermediate radicals on the polymeric chains. Furthermore, the random radical formation and propagation leads to an accelerated abiotic pro-degradation of the investigated PE-based systems.

## 4. Conclusions

Vitamin E and ferulic acid, at high concentrations, i.e., 2 wt.% and 3 wt.%, have been considered as pro-degradant agents for low density polyethylene and all obtained results have been compared to that of neat PE and PE containing the same amount of a classical pro-oxidant agent, such as Ca stearate.

The VE and FA of these high concentrations are able to exert: (i) a slightly pronounced plasticizing action on PE, decreasing slightly the PE rigidity and crystallinity, and increasing the ductility; (ii) a pro-degradant action in PE, promoting the accumulation of oxygen-containing groups and the ductility loss, similarly to the Ca stearate. To sum up, the VE and FA at high concentrations can be considered as suitable eco-friendly pro-oxidant additives for PE, as well as in order to control the polyolefin degradation times. The obtained changes in PE structure, upon the adding of Ca stearate, VE and FA, occur to the same extent; the latter suggests that the pro-oxidant action of the considered compounds cannot be related to some PE morphological variation, which determinates different oxygen penetrations, but it can be related to the pro-degradant ability of these molecules.

## Figures and Tables

**Figure 1 materials-12-00195-f001:**
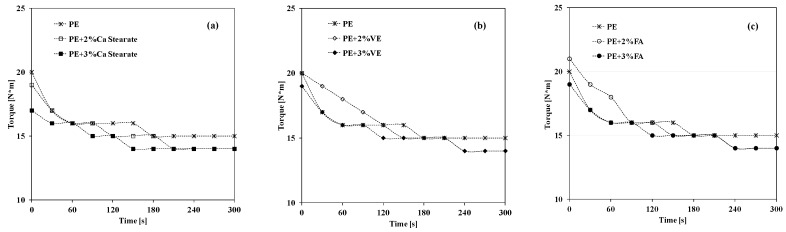
Torque trends of neat PE and PE containing different compounds as a function of the processing time. (**a**) Ca stearate; (**b**) Vitamin E; (**c**) Ferulic Acid.

**Figure 2 materials-12-00195-f002:**
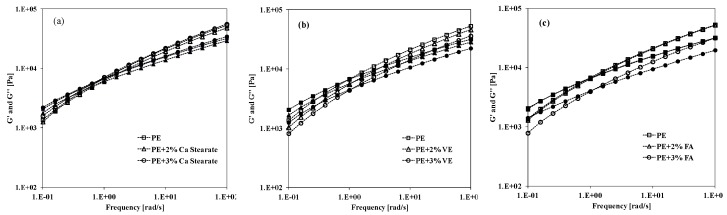
Storage (G′) and Loss (G′′) moduli of neat PE and PE containing different compounds as a function of the frequency. (**a**) Ca stearate; (**b**) Vitamin E; (**c**) Ferulic Acid.

**Figure 3 materials-12-00195-f003:**
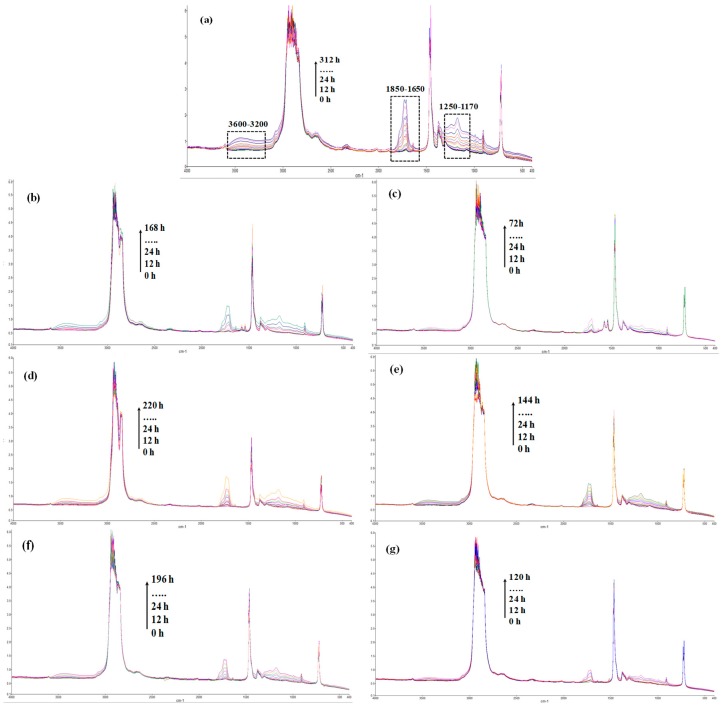
FTIR spectra of neat PE and PE containing different compounds at different photo-oxidation exposure time. (**a**) neat PE; (**b**) 2 wt.% Ca stearate; (**c**) 3 wt.% Ca stearate ;(**d**) 2 wt.% Vitamin E; (**e**) 3 wt.% Vitamin E; (**f**) 2 wt.% Ferulic Acid; (**g**) 3 wt.% Ferulic Acid.

**Figure 4 materials-12-00195-f004:**
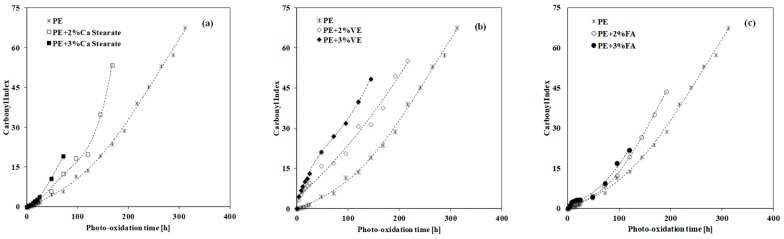
Carbonyl Index of neat PE and PE containing different compounds as a function of the photo-oxidation time. (**a**) Ca stearate; (**b**) Vitamin E; (**c**) Ferulic Acid.

**Figure 5 materials-12-00195-f005:**
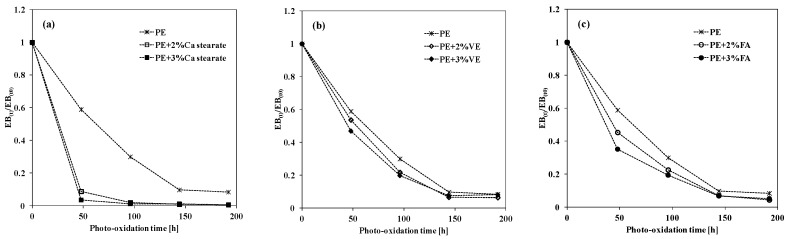
Dimensionless elongational at break EB(t)/EB(t_0_) of neat PE and PE containing different compounds as a function of the photo-oxidation time. (**a**) Ca stearate; (**b**) Vitamin E; (**c**) Ferulic Acid.

**Table 1 materials-12-00195-t001:** Main mechanical properties, i.e., elastic modulus, E, tensile strength, TS, and elongation at break, EB, and fusion (ΔHf) and crystallization (ΔHc) enthalpies for neat PE and PE containing different compounds before UVB exposure.

	E [MPa]	TS [MPa]	EB [%]	ΔH_f_, J/g	ΔH_c_, J/g
PE	215 ± 5	23.3 ± 1.5	578 ± 23	117	−109
PE + 2 wt.% Ca stearate	212 ± 6	22.7 ± 1.2	625 ± 25	109	−93
PE + 3 wt.% Ca stearate	207 ± 7	23.8 ± 1.3	661 ± 27	102	−91
PE + 2 wt.% VE	191 ± 5	23.2 ± 1.3	604 ± 21	105	−98
PE + 3 wt.% VE	177 ± 5	22.4 ± 1.2	626 ± 22	104	−96
PE + 2 wt.% FA	209 ± 6	24.5 ± 1.6	624 ± 23	106	−97
PE + 3 wt.% FA	211 ± 7	21.6 ± 1.3	642 ± 21	102	−101

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
