# Peer review of "Pro-Degradant Activity of Naturally Occurring Compounds on Polyethylene in Accelerate Weathering Conditions"

_materials, 2019, doi:10.3390/ma12010195_

Round 1
Reviewer 1 Report
The paper reports the pro oxidant effet of ferulic acid and vitamin E in case of photo oxidation.
At first, those effects are relatively slight compared to calcium stearate (Figure 5).
The paper would be more easy to understand with some details on the nature of antioxidant package and on the initial OIT value.
About the comment of table 1: what is the link between "plasticizing effect" and a lower crystallinity ? What is the opinion of authors about the increase of mobility of the amorphous phase induced by plasticizer which makes crystallization faster ? It must be better discussed.
A mechanistic explanation of the possible reactions of Vit E and FA which accelerate the oxidation kinetics must be proposed.
What is the meaning of the lines 123-128 ?
Author Response
As notified by the Reviewer, the pro-oxidant effect induced by FA and VE on PE is relatively slight compared to that induced by Ca stearate, but this is an expected result, considering that the Ca stearate is a polymer additive designed to induce pro-oxidation. Moreover, taking into account the natural origin of the FA and VE, the authors believe that the induction of the pro-oxidation on PE in photo-oxidation condition, through the introduction of natural molecules which are able to work in a similar way as a designed additive, can be considered an important issue in this research field. Please, see the new sentence added in the revised manuscript in red font, rows 67-70.
The authors are not reported any OIT values because these values are important for the anti-oxidant activity of these molecules. As known in literature, the naturally occurring molecules at low concentration, i.e. 0.1-0.5 wt.%, in polymers and bio-polymers are able to exert anti-oxidant activity. In this work, the authors shown the ability of these molecules to act as pro-oxidant agent, if added at high concentrations, i.e. 2-3 wt.%.
The adding of some polymer additives having low molecular weight, leads to a significant lowering of the polymer crystallinity, and subsequently the polymer amorphous phase increase. As known, the oxygen penetration and diffusion in the polymer amorphous phase is facilitated with respect to the crystalline one, and for this reason, the amorphous phase usually undergoes faster degradation phenomena.
The adding of 2 and 3 wt.% of FA, VE and Ca stearate leads to a slightly change of the fusion and crystallization enthalpies, in comparison to the neat PE, highlighting a limited influence on the oxygen penetration and diffusion. To explain better this issue, in the end part of paragraph 3.1 have been added new sentences. Please, see the text in red font in the revised manuscript.
A short mechanistic explanation of the possible action of FA and VE in the oxidation accelerating has been proposed. Please, see the text in red font in the end part of paragraph 3.2.
The lines 118-131 come from the template text, we are sorry for the error. During the manuscript preparation, the authors do not erase the original text. In the revised manuscript, the text in lines 118-131 has been canceled. Again, we are sorry for this mistake.
Reviewer 2 Report
Comments:
Introduction:
The explanation why these two compounds were used is highly recommended.
The addition of Vitamin E was studied in other papers - please discuss it.
Materials and methods:
In many places in manuscript there are missed subscripts/ superscripts, i.e. lines: 77,95,99,208 etc. Sometimes the given information can be misleading.
Lines 116-128-shpuld be deleted
Lines 114-116: what was the oxygen concentration and what was the humidity?
Results:
Table 1 - missed tite
Figure 3-missed title
More discussion in aspect of comparison of obtained results with results for other additives used in this kind of polymers is recommended.
References:
No homogeneity in references list, i.e. title of journal: lines 276/277 and 280.
Author Response
In this work, the FA and VE have been considered as natural occurring molecules in order to promote the abiotic pro-oxidation of PE. Please, see the new sentence added in the revised manuscript in red font, rows 67-70.
The lines 118-131 come from the template text, we are sorry for the error. During the manuscript preparation, the authors do not erase the original text. In the revised manuscript, the text in lines 118-131 has been canceled. Again, we are sorry for this mistake.
During the FTIR analysis, the oxygen and humidity are normal atmosphere conditions. This information has been added in the revised manuscript.
The titles of Table 1 and Figure 3 have been added ion the revised manuscript.
The DOI of Ref 5 has been added in the revised manuscript.
The improve the manuscript quality, a short mechanistic explanation of the possible action of FA and VE in the oxidation accelerating has been proposed. Please, see the text in red font in the end part of paragraph 3.2.
Round 2
Reviewer 1 Report
I thanks the authors for their comments. The paper is in part improved.
Some comments however:
1 "The authors are not reported any OIT values because these values are important for the anti-oxidant activity of these molecules."
this sentence is unclear.
Moreover, I feel that it's important to explain of the results presented are valid only for a "pure" PE or in a commercial PE (with a processing stabilization package).
2. About the mechanistic interaction. I am sorry to inform the authors that they are possibly wrong. The prodegrading effect occurs clearly at short times (Figures 4 and 5) contrarily to what they claim and it remains to understand it it comes from a reaction between FA or VA (or their by products) and PE chains or with the stabilizer pacaakage (this is in link with my fabove comment)
Author Response
Based on the current literature, the addition of pro-oxidant agents to the polymers leads to two simultaneous effects, in particular, the formation of new oxygen-containing groups, increasing the carbonyl and/or hydroxyl contents and the fragmentation of the polymer chains, decreasing the molecular weight. In Figure 4, it would seem that the formation of oxygen-containing groups follows the trends: VE > Ca stearate > FA, while, in Figure 5, the trends of EB decay, i.e. the loss ductility, follows: Ca stearate > FA > VE. To sum, the adding of Ca stearate, in comparison to the VE, leads to less pronounced formation of oxygen-containing groups and to an accelerated ductility loss, i.e. chain fragmentation. However, the authors think that it is so hard matter to establish exact relationship between the rate of the formation of oxygen-containing groups and the polymer chain fragmentation and based on this, the use of “pro-oxidant” and “pro-degradant” as synonyms is not exactly always correct. For this reason and to avoid mistake, the authors change the manuscript title from “Pro-oxidant activity ….” to “Pro-degradant activity ….”.
Therefore, the authors agree with the Reviewer that the OIT values could help to understand which PE-based systems is more inclined to from oxygen-containing groups. Really, is this work the authors would report only about the ability of some naturally occurring molecules, such as VE and FA, to act and to be considered as a pro-degradant agents for PE. As known, the action of these molecules as pro-oxidant agent for biological tissue [Ref. 15] and as pro-degradant agent for PLA [Ref. 11] and [Ref. 13], if they are added at high concentration/dose, is notified in the current literature.
Hence, it would seem that the naturally occurring molecules are able to act as pro-degradant agents only at high concentrations, probably, due to the ability of the their radicals to abstract hydrogen atoms form the polymers chains. The formation of the internal radicals onto the macromolecules generally evolves to chain fragmentation, but for some polymers, such as PE, also to crosslinking, as an unwanted side effect. The clarification of this question and some others questions related to a possible reaction between the naturally occurring molecules and the polymer macromolecules will be the scope of future author research works.